# Establishment of a Reference Material in Quality Control for Use in Infectivity and Identity Assays of Recombinant COVID-19 Vaccine, in Accordance with International Standards Organization Guidance

**DOI:** 10.3390/vaccines12090967

**Published:** 2024-08-27

**Authors:** Ana Carolina Ferreira Ballestê Ajorio, Michel Gomes Chagas, Vinicius Pessanha Rhodes, Anderson Peclat Rodrigues, Natália Pedra Gonçalves, Rodrigo Maciel da Costa Godinho, Stephen James Forsythe, Luciana Veloso da Costa, Marcelo Luiz Lima Brandão

**Affiliations:** 1Institute of Technology in Immunobiologicals, Oswaldo Cruz Foundation, Rio de Janeiro 21040-900, Brazil; carolinaajorio@gmail.com (A.C.F.B.A.); michel.chagas@bio.fiocruz.br (M.G.C.); vinicius.rhodes@bio.fiocruz.br (V.P.R.); anderson@bio.fiocruz.br (A.P.R.); natalia.goncalves@bio.fiocruz.br (N.P.G.); rodrigo.godinho@bio.fiocruz.br (R.M.d.C.G.); luciana.costa@bio.fiocruz.br (L.V.d.C.); 2Foodmicrobe.com Ltd., Adams Hill, Keyworth, Nottingham NG12 5GY, UK; sforsythe4j@gmail.com

**Keywords:** recombinant COVID-19 vaccine, infectivity assay, identity assay, quality control, reference material, internal control, pharmaceutical industry, SARS-CoV-2, qPCR

## Abstract

The COVID-19 pandemic, caused by the Severe Acute Respiratory Syndrome Coronavirus 2 (SARS-CoV-2), began in 2019. One of the strategies for pandemic control was mass vaccination. In Brazil, the recombinant COVID-19 vaccine (RCV) was produced on a large scale and offered at no charge to the population. The specifications for quality control analyses of RCV included identity and infectivity determination. To validate the results, a reference material (RM) must be analyzed in parallel with the sample vaccine. This research aimed to establish the RM for use in the identity and infectivity assay for RCV. The candidate RM was analyzed using homogeneity and stability studies. The RM was considered homogeneous for identity (cycle threshold (Ct) ≤ 25.19) and infectivity (average $\overset{-}{x}$ was 9.25 log_10_ infectious units/mL). The RM was considered adequately stable for identity during the total period in all studies, being stable at −70, 5, and 22.5 °C for 380, 313, and 14 days, respectively (Ct ≤ 21.81). For infectivity, the RM was stable at −70, 5, and 22.5 °C for 380, 97, and three days, respectively. Since the property identity and infectivity values of the RM were established, the new RM could be used in quality control analysis.

## 1. Introduction

The pandemic caused by Severe Acute Respiratory Syndrome Coronavirus-2 (SARS-CoV-2) started at the end of 2019. Brazil was among the countries most severely impacted by COVID-19 [1]. According to data from the World Health Organization (WHO), up to 19 December 2023, 35,519,960 cases and 702,116 deaths have been reported [2].

The rapid spread and size of the pandemic posed a challenge for science, pharmaceutical industries, governments, and regulatory agencies. Consequently, it necessitated their rapid mobilization and combined efforts for infection control. The development of effective vaccines was an urgent and highly challenging task for the international scientific community [3].

In Brazil, the Unified Health System (SUS) provides free vaccines to the population, regardless of age, through its National Immunization Program [4,5]. During the pandemic, the Immunobiological Technology Institute (Bio-Manguinhos), a technical–scientific unit of the Oswaldo Cruz Foundation (Fiocruz), was responsible for the production of the recombinant COVID-19 vaccine (RCV). This vaccine was developed through a technology transfer collaboration between AstraZeneca and Oxford University UK [6]. The recombinant ChAdOx1-S/nCoV-19 vaccine is a replication-deficient adenoviral vector vaccine against COVID-19 that expresses the SARS-CoV-2 spike protein gene, instructing the host cells to produce S-antigen protein [7].

Due to the severity of the pandemic in Brazil, large-scale vaccine production was necessary for the potential immunization of the entire population [7]. Production of sterile vaccines, such as RCV, requires the implementation of Good Manufacturing Practices (GMP), ensuring that products are consistently monitored according to appropriate quality standards for their intended use and specified requirements [8].

Two quality control assays for RCV production are infectivity and identity assays, which require a suitable reference material (RM). RM is ‘*a material sufficiently homogeneous and stable with respect to one or more specified properties that has been established to be fit for its intended use in a measurement process*’ [9]. During the RCV assays, AstraZeneca provided initial RM during the technology transfer period, which was analyzed in parallel with each assay to validate the results. However, after the technology transfer period ended, this RM was no longer supplied, and there is no commercially available RM for RCV assays. In this scenario, it was necessary to establish a new RM for quality control assays. This research aimed to establish a new RM from a batch of RCVs.

## 2. Materials and Methods

### 2.1. Batch of RCV to Be Tested as RM for Infectivity and Identity Assays

The RCV batch 212VCD001ZVB was produced by Bio-Manguinhos/Fiocruz and was approved according to the parameters legally defined in the vaccine registration dossier approved by the Brazilian National Health Surveillance Agency. Each ampoule contained ~4 mL of vaccine bulk in glass-sealed vials; viral particle concentration of 1.0 × 10^11^/mL; 0.10% (*w*/*v*) of polysorbate; pH 6.5; ≤0.20 endotoxin units/mL; osmolarity of 421 mOsm/Kg; infectivity of 2.09 × 10^9^ infectious unit (IFU)/mL; sterile; positive for identity test; satisfactory for degree of coloration, integrity, clarity, and opalescence; and DNA to protein ratio of 1.3, and satisfactory aspect.

In order to save on the number of vials used in the studies to establish RM, the same vial was used for both infectivity and identity assays.

### 2.2. Chimpanzee Adenovirus AZD1222 Virus Titration

The titer of the vials of the RM candidate evaluated in this study was determined by the IFU methodology [10]. Two aliquots were used, and serial dilutions (1:10) of the samples were prepared in DMEM medium (Life Technologies, New York, NY, USA) supplemented with 10% of heat-inactivated fetal bovine serum (Life Technologies, New York, NY, USA) and 1% of penicillin–streptomycin (Life Technologies, New York, NY, USA). Then, aliquots (0.1 mL) of the last 3 dilutions were used to inoculate poly-D-lysine-coated 24-well plates containing 0.9 mL of HEK-293 (ATCC^®^ CRL-1573TM) (2.8 × 10^5^ cells/mL). These plates were incubated at 37 °C/5% CO_2_/≥85% relative humidity for 47 h. Non-inoculated wells were used as cell controls. One assay control (AC) lot, C00443-00003, developed and supplied by AstraZeneca during the technology transfer period, was used as RM to validate the assays. AC lower and higher confidentiality limits were previously established in a control chart. After the incubation period, P.A. methanol (J.T.Baker, Trinidad and Tobago) was used to fix the cells, which were then washed with phosphate-buffered saline pH 7.2 (PBS, Sigma-Aldrich, Saint Louis, MO, USA). This was followed by the addition of mouse 1:100 anti-adenovirus (Abcam, Waltham, MA, USA) in each well and maintenance at room temperature for 1 h with light agitation. The liquid was removed, and the wells were washed with PBS. After that, 1:200 rabbit anti-mouse IgG-HRP antibody (Abcam, Waltham, MA, USA) was added to each well. After 1 h at room temperature, the microplates were washed with PBS once more. 1× DAB substrate kit (Thermo Fisher, Rockford, IL, USA) was added in each well, and the plates were kept at room temperature, with slight agitation, for 10 min. Then, UltraPure DNase/RNase-free water (Thermo Fisher, New York, NY, USA) was used to wash the microplates, 1.0 mL of UltraPure water was added to each well of the microplate, and stained cells were counted using an inverted light microscope with a 20 mm field diameter, a 10× ocular lens, and a 10× objective lens (Zeiss, Göttingen, Germany). The titer (log_10_ IFU/mL) was calculated using the Equation (1) below:Titre (log_10_ IFU/mL) = (Average stained cell counts × number of fields × dilution factor)/sample volume(1)

Note that the number of fields in an ocular lens with 20 mm of field diameter, 10× ocular lens, and 10× objective lens is 61, and the sample volume is 0.1 mL.

### 2.3. Identity of the Chimpanzee Adenovirus AZD1222 Virus by Real-Time PCR

The identity of the Chimpanzee adenovirus AZD1222 virus was determined by real-time polymerase chain reaction (qPCR) methodology targeting the flanking region (Flank) and protein spike region (Spike), developed by Oxford University and AstraZeneca and implemented and validated in Bio-Manguinhos during the technology transfer period.

A total of 10 μL of each vaccine sample was diluted in 90 μL (1:10) of sodium dodecyl sulfate (SDS) 0.1% and ethylene diamine tetraacetic acid (EDTA) 25 mM. A Reference Standard’s Code AZD 1222 lot: 122220I, developed and supplied by AstraZeneca, and purified water were used as positive and negative controls, respectively.

Two qPCR reactions were performed, one targeting the Flank and the other Spike region. qPCR Flank mix included: 1× TaqMan Universal Master Mix II (without UNG) (applied biosystems^®^, Vilnius, Lithuania); 12.5 pmol of forward Flank F (5′-AGTCACCGTCCTTGACACG-3′) and reverse Flank R (5′-GAAGCAGAACACAGCACAGG-3′) primers; 6.25 pmol of Reporter Probe 56FAM (5′-TTAATGGAC/ZEN/GCCATGAAGCGAG 3IABkFQ-3′); and 5 μL of the sample (diluted 1000) in a final volume of 25 μL. qPCR Spike mix included: 1X TaqMan Universal Master Mix II (without UNG); 12.5 pmol of forward Spike F (5′-CTGGATCCTCTGAGCGAGAC-3′) and reverse Spike R (5′-TGGTAGATGCCCTTTTCCAC-3′) primers; 6.25 pmol of Reporter Probe 56FAM (5′-AAGTGCACC/ZEN/CTGAAGTCCTTCACC 3IABkFQ-3′); and 5 μL of the sample (diluted 1000) in a final volume of 25 μL. Cycling conditions included DNA Polymerase Activation at 95 °C for 10 min, followed by 35 cycles of 95 °C for 15 s and 60 °C for 45 s in the 7500 Fast System (Applied Biosystems^®^, Singapore). Positive results were considered when the cycle threshold (Ct) values were <30.

### 2.4. Determination of Homogeneity

For the homogeneity study, the titers of 20 randomly selected vials of the RM candidate were determined. The vials were analyzed under duplicate and overrepeatability conditions using the methods described in Section 2.2 and Section 2.3. For the identity assay, the candidate to RM was considered sufficiently homogeneous when all vials gave a positive result in qPCR with Ct < 30 for Spike and Flank targets. For the infectivity assay, the International Harmonized Protocol [11] was used for statistical evaluation of the results. Results were plotted as a graph of results vs. the no. of samples. Then, the graph was visually examined for trends or discontinuities, non-random distribution of differences between the first result and second result, excessive rounding, and intra-sample scattered results. Cochran’s test was used to remove outliers. Then, for homogeneity degree determination, a one-factorial analysis of variances (ANOVA) using 0.25 as the assigned target standard deviation (σp) [12] was applied to the results. To be considered sufficiently homogeneous, the following result must be obtained (Equation (2)):*S^2^sam* < *c*(2)

Note that *S*^2^*sam* = variance between-sample of the batch and *c* = critical value.

For the infectivity assay, which was studied as a quantitative property value, the uncertainty of homogeneity was obtained from the Anova. As the “mean squared between units of the batch” (MS_between_) of the candidate RM was greater than the “mean squared within units of the batch” (MS_within_), the following Equation (3) was applied [13]:u (homogeneity study) = √((MS_between_ − MS_within_)/*n*)(3)

Note, *n* = number of replicates used in the assay.

### 2.5. Determination of Stability

Individual vials were analyzed for each combination of temperature and duration. The method of analysis is described in previous Section 2.2 and Section 2.3, and the stability estimation was realized according to ISO [13].

A long-term stability study at different storage temperatures (−70 ± 10 °C and 5 ± 3 °C) was conducted for 380 and 313 days, respectively, using the classical approach. At −70 ± 10 °C, two vials were evaluated on days 0, 7, 14, 21, 28, 45, 62, 153, 243, 313, and 380, giving a total of 22 analyses. At 5 ± 3 °C, aliquots from the same two vials were evaluated on days 0, 1, 2, 3, 7, 8, 9, 14, 15, 21, 22, 28, 45, 62, 153, 243, and 313, totaling 34 determinations.

A short-term stability study at 22.5 ± 2.5 °C, simulating room temperature, was performed for 66 days. Two items were analyzed on days 0, 1, 3, 7, 8, 9, and 14, totaling 14 analyses. The results were analyzed for the usual indications in diagnostics (non-random distribution of differences between items, trends or discontinuities, and excessive rounding). Then, regression residual analysis was applied to determine if the linear regression of the property values revealed any trend over time [13].

The infectivity assay was assessed as a quantitative property value. The long-term stability uncertainty at −70 ± 10 °C was calculated using Equation (4):u(stability) = Sb × *t*(4)

Note, Sb = error of the slope; *t* = no. of days

### 2.6. Certified Value Metrological Traceability

Certified value traceability was verified by (a) the batch dossier of production, which consists of a description of the source of the raw materials and the history of their production; (b) the use of the AC lot C00443-00003 and Reference Standard AZD 1222 lot 122220I developed and supplied by AstraZeneca as the RM to validate the assays infectivity and identity assay, respectively; and (c) data acquisition was registered, following protocols that were approved previously and followed the GMP requirements. The infectivity and identity methods used in the present study were validated, the equipment and instruments were calibrated, and results were generated under controlled conditions to guarantee reliability.

## 3. Results

### 3.1. Homogeneity Assessment

The results of the analyses of the homogeneity study are presented in Table 1.

As all vials showed Ct < 30 for both replicates, the candidate RM was considered sufficiently homogenous for this property value identity. Regarding infectivity, no outliers’ values were identified using the Cochran test. The concentration average was 9.25 ± 0.025 log_10_ IFU/mL, considered sufficiently homogeneous. The result for the homogeneity accuracy was valid, as its criteria were met: the critical *c* value (0.0094) > *S*^2^*am* (0.00065).

### 3.2. Stability Assessment

Data obtained in the stability studies are presented in Table 2. As all vials showed Ct < 30 for all combinations of temperatures and periods of time, the candidate RM was considered adequately stable for the property value identity at −70 ± 10 °C, 5 ± 3 °C, and 22.5 ± 2.5 °C for 380, 313, and 14 days, respectively.

Table 3 describes the linear regression analysis of the stability data of the infectivity assay. The candidate RM was considered stable for the period of the study only at −70 ± 10 °C. The calculated uncertainty value was 0.152 IU/mL. At 5 ± 3 °C and 22.5 ± 2.5 °C, the trend analysis of the graph (Figure 1 and Figure 2) and linear regression analysis (Table 3) showed a negative slope. This indicated some instability in the infection at these temperatures. However, the candidate RM was considered sufficiently stable over 97 days and 3 days at 5 ± 3 °C and 22.5 ± 2.5 °C, respectively.

## 4. Discussion

To establish an RM, the first requirement is a homogeneity assessment [13]. In the case of vaccines, the candidate homogeneity can be evaluated by an inter-laboratory study, generally including laboratories that are part of the vaccine producers or organizations and/or government laboratories responsible for quality control [12,14,15]. For this approach, ISO Standards concerning RM must be followed, which aim to standardize the requirements for RM establishment [13,16]. In this study, the candidate RM was considered homogeneous according to the statistical analysis applied [11] for the infectivity assay, which used a quantitative approach. Previously, this protocol has been used for the homogeneity evaluation of RM for bacteria in foods [17,18] and viruses in vaccines [12]. Ajório et al. [12] produced a certified RM sufficiently homogeneous with a concentration average of 4.68 log_10_ International Units/human dose, also applying an assigned target σ_p_ of 0.25. For the identity assay, the most restricted criteria were applied, and the candidate RM was only considerably sufficiently homogeneous because all vials presented positive results in qPCR. qPCR was applied by Kempster et al. [19] to assess the homogeneity of a Zika virus nucleic acid standard using a relative power assay compared with an international standard. Other techniques are also applied to verify the identity of viruses, which generally are based on enzyme-linked immunosorbent assays [20,21].

Property value stability is essential for an RM establishment. At the moment of use, the property value should be consistent with the value stated in the RM documentation [13]. At the reference temperature, −70 ± 10 °C, during the entire study period (380 days), the RM was stable and showed 1.6 × 10^−4^ of slope per day (Table 3). This linearity was also observed in Figure 1, demonstrating that the RM is expected to remain stable in this condition for a longer time period. Similar results were observed by Penaud-Budloo et al. [20], who studied the stability of an adeno-associated virus 8 reference standard for 2 years at ≤−70 °C and observed that the potency remained stable. Ajório et al. [12] produced a certified RM containing yellow fever virus stable at −20 °C for 715 days with a slope per day value of 6.6 × 10^−6^. Perraut et al. [22] performed a stability study with yellow fever vaccines for up to 1080 days at −20 °C and observed that the vaccine maintained its potency according to specification and would maintain its stability for a much longer period.

At a refrigeration temperature of 5 ± 3 °C, the RM could be maintained for 97 days (~3 months), a period considered satisfactory to send the RM to other laboratories for storage in a refrigerator until required. To keep it for a longer time period, the laboratory could freeze it, but infectivity variation due to defrosting and freezing again was not evaluated in the present study. The short-term study at 22.5 ± 2.5 °C for 14 days demonstrated that the RM was sufficiently stable for only 3 days (Table 3). Significant loss (0.14 log_10_ IFU/mL) after 7 days was observed (Figure 2). The study of this storage temperature was intended to consider possible situations, such as the event of a storage equipment (freezer or refrigerator) breakdown. Another application is transporting the material to other facilities without refrigeration. The latter might occur in order to reduce transportation costs [17,18,23]. Ajório et al. [12] reported that the yellow fever virus vaccine was sufficiently stable at 22.5 °C for 183 days. This prolonged stable period is because it is a freeze-dried vaccine. When the authors studied the stability of the yellow fever vaccine after its reconstitution with water at 5 °C for 3 days, the infectivity was not sufficiently stable [12].

Unfortunately, due to the lack of laboratories in Brazil carrying out this test, it was not possible to perform a collaborative study. International laboratories from other pharmaceutical companies were not available to participate in this study. Consequently, it was not possible to certify the RM.

## 5. Conclusions

The recombinant COVID-19 vaccine lot 212VCD001ZVB was established as an RM for the values of identity and infectivity (infectivity by IFU methodology). After establishing the RM for infectivity and a quantitative property value, 30 assays were performed to develop a control chart to determine the confidence limits. The RM can be used in the analysis of infectivity and identity assays when stored at −70 ± 10 °C and when stored at 5 ± 3 °C for 97 days since its properties were stable during this time period.

## Figures and Tables

**Figure 1 vaccines-12-00967-f001:**
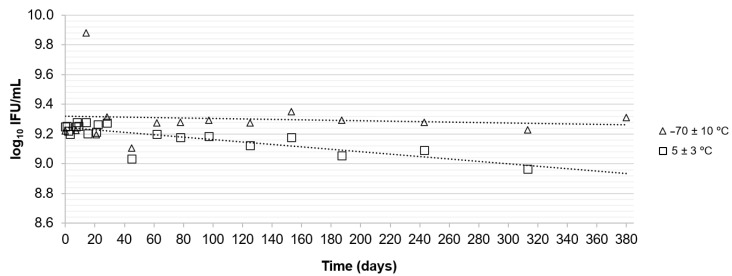
Results of the long-term stability study at (△) −70 ± 10 °C and (☐) 5 ± 3 °C over a period of 380 and 313 days, respectively. Each point corresponds to the mean count of two candidate reference material vials in log_10_ infectious units/mL.

**Figure 2 vaccines-12-00967-f002:**
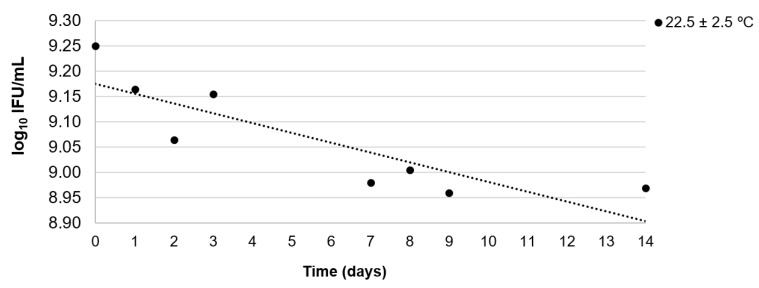
Results of the short-term stability study at 22.5 ± 2.5 °C. Each point corresponds to the mean count of two candidate reference material vials in log_10_ infectious units/human dose.

**Table 1 vaccines-12-00967-t001:** Analysis of the 20 vials used in the homogeneity study.

Vial	Infectivity Assay (log_10_ IFU ^1^/mL)		Identity Assay (Ct ^2^)			
			Flank		Spike	
	Replicate 1	Replicate 2	Replicate 1	Replicate 2	Replicate 1	Replicate 2
1	9.31	9.31	22.99	21.52	20.70	20.42
2	9.27	9.26	23.40	20.87	21.93	20.11
3	9.24	9.22	22.60	21.25	21.65	20.78
4	9.24	9.29	21.66	23.07	20.95	21.36
5	9.22	9.25	22.87	22.16	23.76	20.51
6	9.30	9.20	24.70	22.39	25.19	20.55
7	9.30	9.25	23.70	22.19	22.76	19.65
8	9.28	9.31	22.31	21.35	20.39	20.63
9	9.21	9.22	22.20	22.19	20.97	20.20
10	9.24	9.25	24.91	20.97	19.66	19.94
11	9.30	9.33	19.62	22.11	20.10	20.99
12	9.25	9.26	19.67	21.38	19.71	20.51
13	9.26	9.18	19.49	21.36	20.02	18.71
14	9.19	9.20	19.73	20.51	19.53	19.65
15	9.26	9.29	20.60	21.64	20.44	20.04
16	9.26	9.23	20.04	22.01	20.38	21.11
17	9.20	9.24	19.89	21.66	19.80	20.58
18	9.19	9.21	21.17	22.07	21.07	20.14
19	9.25	9.26	20.92	23.86	20.59	19.49
20	9.21	9.27	21.89	21.88	21.13	19.02

^1^—infectious unit; ^2^—cycle threshold.

**Table 2 vaccines-12-00967-t002:** Data from the stability studies of the candidate reference material.

Time (Days)	Temperature (°C)								
	−70 ± 10			5 ± 3			22.5 ± 2.5		
	Infectivity ^1^	Flank ^2^	Spike ^2^	Infectivity	Flank	Spike	Infectivity	Flank	Spike
0	9.23	21.26	20.57	9.25	20.92	19.83	9.25	21.37	20.45
1	NR ^3^	NR	NR	9.24	21.69	20.68	9.16	21.56	20.60
2	NR	NR	NR	9.22	20.56	20.10	9.06	20.44	20.25
3	NR	NR	NR	9.19	20.68	20.28	9.12	20.10	19.72
7	9.23	20.56	19.43	9.25	20.53	19.62	8.98	20.48	19.98
8	NR	NR	NR	9.28	20.75	20.07	9.00	20.48	19.42
9	NR	NR	NR	9.25	20.67	19.90	8.96	20.43	19.56
14	NR	19.73	18.70	9.28	19.87	19.38	8.97	19.99	19.19
21	9.20	21.38	20.32	9.21	21.36	20.54	NR	NR	NR
28	9.32	20.78	19.77	9.27	20.67	19.91	NR	NR	NR
45	9.11	20.04	18.87	9.03	20.13	19.32	NR	NR	NR
62	9.28	20.64	19.49	9.20	20.69	19.99	NR	NR	NR
153	9.35	21.81	20.66	9.18	21.06	20.13	NR	NR	NR
243	9.28	17.99	17.87	9.09	18.39	18.00	NR	NR	NR
313	9.23	20.95	19.08	8.96	21.04	20.25	NR	NR	NR
380	9.31	21.50	20.45	NR	NR	NR	NR	NR	NR

^1^—Results expressed in infectious units/mL; ^2^—Results expressed in cycle threshold; ^3^—Not realized.

**Table 3 vaccines-12-00967-t003:** Results of the statistical analysis of the stability studies of the infectivity of the candidate reference material according to ISO Guide 35:2017 [13].

Storage Temperature	Type of Study	Period (Days)	log_10_ IFU ^1^/mL		95% Confidence Interval		Result
			Slope Per Day	Standard Error Per Day	Lower	Higher	
−70 ± 10 °C	Long-term	380	−0.00016	0.00040	−0.0010	0.00069	Sufficiently stable
5 ± 3 °C	Long-term	313	−0.00081	0.00013	−0.0011	−0.00053	Insufficiently stable
		97	−0.00094	0.00049	−0.0020	0.00011	Sufficiently stable
22.5 ± 2.5 °C	Short-term	14	−0.018	0.005	−0.030	−0.007	Insufficiently stable
		3	−0.048	0.026	−0.16	0.064	Sufficiently stable

^1^—infectious units.

## Data Availability

Data are contained within the article.

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
