# Peer review of "Establishment of a Reference Material in Quality Control for Use in Infectivity and Identity Assays of Recombinant COVID-19 Vaccine, in Accordance with International Standards Organization Guidance"

_vaccines, 2024, doi:10.3390/vaccines12090967_

Round 1
Reviewer 1 Report
Comments and Suggestions for Authors
1. What is the main question addressed by the research?
In the present study, Ajorio et al tried to establish a certified reference material (RM) to use as an internal/quality control for the large-scale production of the recombinant COVID-19 vaccine (RCV). This involves ensuring the reference material homogeneity in terms of infectivity and identity, and stability to validate its effectiveness in quality control assay for RCV.
2. What parts do you consider original or relevant for the field? What
the specific gap in the field does the paper address?
Parts that are original or relevant for the field-
Establishment of Certified Reference Material (RM): The study focuses on establishing certified RM as an internal control for the production of RCV. This is key for ensuring consistent and reliable quality control in the large-scale production of RCV.
Infectivity and Identity Testing: The study tested the homogeneity of the RM in terms of identity and infectivity under various conditions.
Stability Testing: The study tested the stability of the reference material at different temperatures including both low and ambient room temperature for different days.
Overall, this paper addresses the gap in the availability of certified RM for the potency assay of recombinant COVID-19 vaccine (RCV). The lack of certified RM can lead to inconsistencies and inaccuracies in the quality control process of large-scale vaccine production. By establishing a certified RM, the study provides a standardized and reliable internal control that can be used in the routine analysis of RCV, thereby improving the overall quality, efficiency, and safety of the vaccine.
3. What does it add to the subject area compared with other published
material?
Most of the previous studies on reference materials lack comprehensive testing across multiple parameters (e.g., homogeneity, and stability). This study utilizes a comprehensive approach including homogeneity and stability ensuring RM's effectiveness.
Stability testing in many previous studies may not cover a wide range of temperatures or time periods. This study conducts stability testing at various temperatures and on different days. The authors claimed that the RM remains stable at −70, 5, and 22.5 ºC for 380, 97, and 3 days respectively. Thus, this study offers significance in the field by establishing a certified RM for RCV.
4. What specific improvements should the authors consider regarding the
methodology? What further controls should be considered?
Additional Temperature Range for Stability Testing: In the present study authors checked the stability of RM at −70, 22.5, and 5 ºC. The inclusion of additional temperatures such as -20 ºC and 37 ºC can give a more comprehensive knowledge about stability.
Extended Time Points for Stability Studies: In the current study, stability of RM at different temperatures were tested for up to 380, 313 days. An extension of the study duration beyond 380 days to evaluate long-term stability can be useful.
Cross-laboratory validation: No cross-laboratory validation was performed in this study. This is essential to ensure RM reliability and potency in different settings.
Addition of Functional Assays: This study mainly focuses on homogeneity, infectivity, and stability. The inclusion of functional assays such as PRNT, immunogenicity, antigenicity, or neutralization assay will ensure that the RM not only maintains stability but also retains its biological function at different temperatures and periods tested.
Further control to consider-
Cross-laboratory Validation: Expand the validation study to include multiple independent laboratories across different geographic locations. This will ensure the RM's reliability in various settings.
Comparison with Other Reference Standards: Compare the new RM with other existing reference standards from different manufacturers. This will compare the new RM against established materials and provide additional validation.
Aliquots stability: Did the author check if small aliquots of RM are stable?
The effect of other adverse conditions on RM stability can be beneficial.
5. Please describe how the conclusions are or are not consistent with the
evidence and arguments presented. Please also indicate if all main questions
posed were addressed and by which specific experiments.
The conclusions described here are consistent with the evidence provided. For example-
Infectivity and identity: The conclusion that the RM is identical and infective is addressed by infecting the cells with RM and determining the Ct value of viral particles in different aliquots.
The conclusion that RM is stable for use in quality control assays is addressed by testing stability at different temperatures and different times.
6. Are the references appropriate?
The references used in this manuscript are appropriate, relevant, and effectively supporting the study objectives. Some of the additional references that can be considered are-; https://doi.org/10.1038/s41392-021-00621-4
7. Please include any additional comments on the tables and figures and
quality of the data.
Figures, tables, and data quality look good. Some minor suggestions are-
1. Page 2, line 54-The production of sterile medicines, such as the RCV. RCV is not medicine. Please replace ‘medicines’ with ‘vaccine’.
2. Page 2, line 86- ‘realized’ should be ‘released’
3. Page 3, lines 112-115- Please provide references.
4. Page 3, line 133-Singopore in place of singapire
5. Please discuss if any other reference material is available for RCV. If yes, how the RM described in this paper is better than the previously discovered one?
Comments on the Quality of English LanguageThe quality of the English language in the present manuscript is good. Only minor correction is needed.
Author Response
Comments 1: Figures, tables, and data quality look good. Some minor suggestions are:
Page 2, line 54 - The production of sterile medicines, such as the RCV. RCV is not medicine. Please replace ‘medicines’ with ‘vaccine’.
Response 1: The text was corrected as suggested (L58).
Comments 2: Page 2, line 86- ‘realized’ should be ‘released’
Response 2: The text was modified (L93).
Comments 3: Page 3, lines 112-115- Please provide references.
Response 3: This method was developed by the Oxford University and AstraZeneca and is not described in any Pharmacopeia or other official compendium. We are describing this method in the literature in the first time. So, there is not a reference to be cited.
Comments 4: Page 3, line 133-Singopore in place of singapire
Response 4: The word was corrected as suggested (L146).
Comments 5: Please discuss if any other reference material is available for RCV. If yes, how the RM described in this paper is better than the previously discovered one?
Response 5: There is no other RM available for RCV. This information was included in the text (L70).
Reviewer 2 Report
Comments and Suggestions for Authors
- Clarity and Precision: Ensure the terminology is consistent and precise (e.g., "Severe Acute Respiratory Syndrome Coronavirus-2" instead of "Serious Intense Respiratory Syndrome Coronavirus-2").
- Grammar and Style: Adjust sentence structures for better readability and flow.
- Data Presentation: Clearly present the data with appropriate use of scientific notation and units.
- Acronym Consistency: Use acronyms consistently after their first introduction (e.g., SARS-CoV-2, RCV, RM).
- Formal Tone: Maintain a formal and scientific tone throughout the summary.
MATERIALS AND METHODS:
REWRITE AND MAKE GRAPHICAL ABSTRACT (SUGGESTION TO USE BIORENDER). THE METHODS ARE CONFUSED.
Suggestions for Improving Tables and Graphs in RStudio
-
Data Presentation:
- Use Clear Titles and Labels: Ensure each table and graph has a clear, descriptive title and labels for all axes and legends.
- Consistency: Maintain consistent formatting, including font type, size, and color schemes, across all visualizations for a cohesive look.
-
Enhanced Visual Appeal:
- Color Palettes: Utilize color palettes that are both visually appealing and accessible, ensuring sufficient contrast for readability. Tools like RColorBrewer or viridis can be helpful.
- Themes: Apply themes (e.g., theme_minimal(), theme_bw()) from ggplot2 to improve the aesthetic quality of your plots.
-
Data Clarity:
- Avoid Overcrowding: If a graph has too many data points, consider summarizing the data or using interactive plots (e.g., with plotly) where users can zoom and hover for details.
- Annotations: Use annotations to highlight key data points or trends within the graphs.
-
RStudio Tools:
- ggplot2: For advanced plotting capabilities, including faceting, layering, and customization.
- plotly: To create interactive and dynamic plots that enhance user engagement.
- shiny: For building interactive web applications that can incorporate real-time data visualization.
- knitr and rmarkdown: For integrating tables and plots into dynamic reports, ensuring that analyses are reproducible and well-documented.
-
Data Summary Tables:
- kable or gt: Use these packages to create well-formatted tables that can be easily embedded into R Markdown documents.
- Conditional Formatting: Highlight important values or changes using conditional formatting techniques available in gt.
POINTS TO CHANGE IN THE TEXT:
Brazil was among the countries most severely impacted by COVID-19
The pandemic dynamics posed significant challenges for science, pharmaceutical industries, governments, and regulatory agencies. Rapid mobilization and combined efforts were essential to develop the necessary tools to combat the pandemic.
Developing effective vaccines was an urgent and highly challenging task for the international scientific community.
In Brazil, the Unified Health System (SUS) provides free vaccines to the population through its National Immunization Program
During the pandemic, the Immunobiological Technology Institute (Bio-Manguinhos), a technical-scientific unit of the Oswaldo Cruz Foundation (Fiocruz), was responsible for producing the recombinant COVID-19 vaccine (RCV). This vaccine was developed through a technology transfer with AstraZeneca and Oxford University, UK
The ChAdOx1-S/nCoV-19 [recombinant] vaccine is a replication-deficient adenoviral vector vaccine against COVID-19 that expresses the SARS-CoV-2 spike protein gene, instructing host cells to produce the S-antigen protein.
Due to the severity of the pandemic in Brazil, large-scale vaccine production was necessary to immunize the entire population.
Producing sterile medicines, such as RCV, requires the implementation of Good Manufacturing Practices (GMP), ensuring that products are consistently monitored according to appropriate quality standards for their intended use and specified requirements.
The RCV production specifications included infectivity and identity assays, requiring suitable reference material (RM). RM is defined as ‘a material sufficiently homogeneous and stable with respect to one or more specified properties, established to be fit for its intended use in a measurement process’
During the RCV assays, AstraZeneca provided initial RM during the technology transfer period, which was analyzed in parallel with each assay to validate the results. However, after the technology transfer period ended, this RM was no longer supplied.
In this scenario, it was necessary to establish new RM for quality control assays. This study aimed to establish new RM from a batch of RCV to be used as control material during identity and infectivity assays in the RCV production chain
Comments on the Quality of English LanguageEXTENSIVE REVIEW THE ENGLISH BY NATIVE.
Author Response
Comments 1: Clarity and Precision: Ensure the terminology is consistent and precise (e.g., "Severe Acute Respiratory Syndrome Coronavirus-2" instead of "Serious Intense Respiratory Syndrome Coronavirus-2").
Response 1: The term was modified as suggested (L35).
Comments 2: Grammar and Style: Adjust sentence structures for better readability and flow.
Response 2: The grammar and style were revised by the English native co-author Prof. Stephen Forsythe.
Comments 3: Data Presentation: Clearly present the data with appropriate use of scientific notation and units.
Response 3: The data, including the scientific notation and units were revised as suggested.
Comments 4: Acronym Consistency: Use acronyms consistently after their first introduction (e.g., SARS-CoV-2, RCV, RM).
Response 4: The acronyms were corrected in all text as suggested.
Comments 5: Formal Tone: Maintain a formal and scientific tone throughout the summary.
Response 5: The text was revised for formal and scientific tone throughout the summary by the English native co-author Prof. Stephen Forsythe.
Comments 6: Rewrite and make graphical abstract (suggestion to use Biorender). The methods are confused.
Response 6: The graphical abstract was rewritten and the images used were from Biorender.
Comments 7: Brazil was among the countries most severely impacted by COVID-19.
Response 7: The text was modified as suggested (L36-37).
Comments 8: The pandemic dynamics posed significant challenges for science, pharmaceutical industries, governments, and regulatory agencies. Rapid mobilization and combined efforts were essential to develop the necessary tools to combat the pandemic. Developing effective vaccines was an urgent and highly challenging task for the international scientific community. In Brazil, the Unified Health System (SUS) provides free vaccines to the population through its National Immunization Program. During the pandemic, the Immunobiological Technology Institute (Bio-Manguinhos), a technical-scientific unit of the Oswaldo Cruz Foundation (Fiocruz), was responsible for producing the recombinant COVID-19 vaccine (RCV). This vaccine was developed through a technology transfer agreement with AstraZeneca and Oxford University, UK. The ChAdOx1-S/nCoV-19 [recombinant] vaccine is a replication-deficient adenoviral vector vaccine against COVID-19 that expresses the SARS-CoV-2 spike protein gene, instructing host cells to produce the S-antigen protein. Due to the severity of the pandemic in Brazil, large-scale vaccine production was necessary to immunize the entire population. Producing sterile medicines, such as RCV, requires the implementation of Good Manufacturing Practices (GMP), ensuring that products are consistently monitored according to appropriate quality standards for their intended use and specified requirements. The RCV production specifications included infectivity and identity assays, requiring suitable reference material (RM). RM is defined as ‘a material sufficiently homogeneous and stable with respect to one or more specified properties, established to be fit for its intended use in a measurement process’. During the RCV assays, AstraZeneca provided initial RM during the technology transfer period, which was analyzed in parallel with each assay to validate the results. However, after the technology transfer period ended, this RM was no longer supplied. In this scenario, it was necessary to establish new RM for quality control assays. This study aimed to establish new RM from a batch of RCV to be used as control material during identity and infectivity assays in the RCV production chain.
Response 8: The text was modified as suggested (L34-74)
Reviewer 3 Report
Comments and Suggestions for Authors
Thank you for sharing your manuscript on the establishment of reference material for quality control to be used in infectivity and identity assays of a rCOVID-19 vaccine aligned with guidance of the International Standards Organisation. The following comments may help to improve the article:
L39-40: The sentence "It was necessary, [...] to face the pandemic." reads a bit rocky. Please revise.
L43-44: To the entire population disregarding of e.g. age? Please clarify in your manuscript.
L47-48: Using "through, due to" does not read well. Please revise.
L58-59: The use of "included" and "includes" in this context is confusing. Please reconsider whether past or present should be used here and revise accordingly.
L59, L61: "is'" and "process'"please remove ' in the context.
L62: You introduced the meaning of RM already in L59; please revise.
L64: Validate or possibly validation? Please revise.
L74: "vaccine bulk in glass sealed vials" reads a bit rocky. Please revise.
L74-78: Is the intention of the information proved to describe properties of an ampoule? Please revise for better readability.
L76: infectivity (infectivity) seems duplicated.
L85-86: "After, inoculate [...] were realised [...]" reads somewhat rocky. Please revise.
L88-89: Please provide more information on the controls.
L92: I may have missed it, but which concentration did PBS have?
L84: Please proved products information of DMEM and FBS.
L91: Pure methanol or any other concentration?
L97: At which temperature was the incubation performed?
L93, L96: At which concentration were anti-adenovirus and anti-mouse IgG-HRP added?
L116: Which sample are your referring to? Could it be from the qPCR reaction? Please revise.
L119: Dilution of the positive control? Please revise.
L112-134: Please revise the entire section as it is really hard to follow.
General comment: Regarding explanations on abbreviations used in tables, please check the journal's guidelines whether footnotes are wanted.
L233-238 and L259-262: Lengthy sentences, please revise.
Comments on the Quality of English Language
I have highlighted some sections in the manuscript that need revision. Please consult an English editing service to improve the overall readability of the manuscript.
Author Response
Comments 1: L39-40: The sentence "It was necessary, [...] to face the pandemic." reads a bit rocky. Please revise.
Response 1: The text was modified as suggested (L43).
Comments 2: L43-44: To the entire population disregarding of e.g. age? Please clarify in your manuscript.
Response 2: Yes, independently of age. The text was modified as suggested (L47).
Comments 3: L47-48: Using "through, due to" does not read well. Please revise.
Response 3: The text was modified as suggested (L47).
Comments 4: L58-59: The use of "included" and "includes" in this context is confusing. Please reconsider whether past or present should be used here and revise accordingly.
Response 4: The word was modified as suggested (L62, L63).
Comments 5: L59, L61: "is'" and "process'" please remove ' in the context.
Response 5: This definition is reproduced as given in the ISO Guide 30:2017. We think this exact copying should be clearly indicated (L64-65).
Comments 6: L62: You introduced the meaning of RM already in L59; please revise.
Response 6: The sentence was revised as suggested.
Comments 7: L64: Validate or possibly validation? Please revise.
Response 7: The word validate is correct. With during the assay, the RM shows the results according to control chart specifications, the assay is considerate valid (L68).
Comments 8: L74: "vaccine bulk in glass sealed vials" reads a bit rocky. Please revise.
Response 8: These terms are correct and are commonly used in the pharmaceutical industry.
Comments 9: L74-78: Is the intention of the information proved to describe properties of an ampoule? Please revise for better readability.
Response 9: The aim was to present the results obtained in the tests as required for the batch to be released by quality control.
Comments 10: L76: infectivity (infectivity) seems duplicated (L82).
Response 10: The duplicated word was removed was suggested.
Comments 11: L85-86: "After, inoculate [...] were realised [...]" reads somewhat rocky. Please revise.
Response 11: The sentence was revised as suggested (L93-94).
Comments 12: L88-89: Please provide more information on the controls.
Response 12: More information has been provided as suggested (L98-99).
Comments 13: L92: I may have missed it, but which concentration did PBS have?
Response 13: The pH is 7.2 and salt concentration according to specification is “9.07-10.0 g/L”.
Comments 14: L84: Please proved products information of DMEM and FBS.
Response 14: The information is now provided as suggested (L90-91).
Comments 15: L91: Pure methanol or any other concentration?
Response 15: Pure methanol was used. This information is now provided as suggested (L100).
Comments 16: L97: At which temperature was the incubation performed?
Response 16: Room temperature. This information is given in the text as suggested (L108).
Comments 17: L93, L96: At which concentration were anti-adenovirus and anti-mouse IgG-HRP added?
Response 17: Anti-adenovirus 1:100 in PBS and anti-mouse IgG-HRP 1:200 in PBS. The information is provided as suggested (L102 / L105).
Comments 18: L116: Which sample are your referring to? Could it be from the qPCR reaction? Please revise.
Response 18: The vaccine sample. This information is given in the text as suggested (L127).
Comments 19: L119: Dilution of the positive control? Please revise.
Response 19: The sentence was removed (L130).
Comments 20: L112-134: Please revise the entire section as it is really hard to follow.
Response 20: The section was revised as suggested.
Comments 21: General comment: Regarding explanations on abbreviations used in tables, please check the journal's guidelines whether footnotes are wanted.
Response 21: All the footnotes are provided in the Tables according to journal's guidelines.
Comments 22: L233-238 and L259-262: Lengthy sentences, please revise.
Response 22: These sentences were revised as suggested. Results and Discussion sections were separated.
Round 2
Reviewer 3 Report
Comments and Suggestions for Authors
Thank you for addressing most of my comments.